# Fluorescent Azasteroids through Ultrasound Assisted Cycloaddition Reactions

**DOI:** 10.3390/molecules26165098

**Published:** 2021-08-23

**Authors:** Costel Moldoveanu, Ionel Mangalagiu, Gheorghita Zbancioc

**Affiliations:** 1Chemistry Department, Alexandru Ioan Cuza University of Iasi, 11 Carol 1st Bvd, 700506 Iasi, Romania; ionelm@uaic.ro; 2Institute of Interdisciplinary Research-CERNESIM Centre, Alexandru Ioan Cuza University of Iasi, 11 Carol I, 700506 Iasi, Romania

**Keywords:** azasteroid, benzo[f]quinoline, cycloaddition, ultrasound irradiation, fluorescent

## Abstract

We report here the synthesis and optical spectral properties of several new azasteroid derivatives. The formation of these compounds was explained based on the most probable mechanism. The luminescent heterocycles were synthesized by 1,3-dipolar cycloaddition reactions between benzo[f]quinoline and methylpropiolate or dimethyl acetylenedicarboxylate (DMAD). A selective and efficient way for [3+2]-dipolar cycloaddition of benzo[f]quinolinium ylides under ultrasound (US) irradiation (20 kHz processing frequency) is presented. We report substantially higher yields under US irradiation, whereas the solvent amounts required are at least three-fold less compared to classical heating. The azasteroid derivatives are blue emitters with λ_max_ of fluorescence around 430–450 nm. A certain influence of the azasteroid substituents concerning absorption and fluorescent properties was observed. Compounds anchored with a bulky pivaloyl group or without a C=O carbonyl group have shown increased fluorescence intensity.

## 1. Introduction

Azasteroids are an important class of heterocycles that has received increasing interest in recent years [1], due to a wide range of potential applications for medicinal chemistry and optoelectronics. As materials with potential applications in medicinal chemistry, azasteroid derivatives were investigated as potential antimicrobial [2], antifungal [3], anticancer [4,5], and antituberculosis [6] agents. The optoelectronic properties of the azasteroids were investigated regarding their fluorescence emission [7,8,9]. The fluorescence of the azasteroid derivatives makes them very attractive materials in optoelectronics, whereas a combined use of the two distinct properties (biological and optical) has suggested interesting applications as fluorescent biomarkers [10,11,12,13].

Azasteroid derivatives are in fact benzo-azaindolizines, a class with an 18 π-electron N-fused heterocycle, containing a bridgehead nitrogen atom shared by an electron-excessive pyrrole and an azine electron-deficient six-membered ring. This structural arrangement being a ‘pure’ blue-emitting moiety. [13,14,15,16,17]. This uneven π-electron distribution between the two fused rings is an important feature that leads to electron delocalization. The electron delocalization within the entire heterocycle skeleton can be possible without a planar geometry of the indolizine. The planarity of azasteroid derivatives is provided by the sp^2^ hybridization of all the atoms in the fused ring and is preserved upon substitution with different groups.

In recent years, ultrasound-assisted reactions have proved to be a new versatile tool in synthetic organic chemistry [18,19], offering a facile and usefully alternative in a large variety of syntheses [20,21,22,23,24,25,26,27]. In comparison with conventional thermal heating (TH), ultrasound irradiation has several important advantages in terms of higher yields, increased purity of the compounds and selectivity, shorter reaction times, lower costs, and simplicity in handling and processing. All these advantages make US-assisted reactions environmentally friendly [22,23,24,28].

Our group has presented in previous work several studies on the synthesis of azine derivatives and their biological activities [29,30,31,32,33,34]. A special focusing was paid to the dipolar cycloaddition reactions of azinium ylides with various dipolarophiles in order to obtain new compounds with blue-fluorescent properties. Herein, we add further contributions in the field of using US irradiation in organic chemistry [20,21,22,23,24] and propose a facile, efficient, and environmentally friendly method for the synthesis of blue-emitting derivatives using US technologies.

## 2. Results and Discussion

Scheme 1 presents the general strategy adopted for the synthesis of the new fluorescent azasteroids. As can be observed, the synthesis of all pyrrolobenzo[f]quinoline derivatives, **5a**–**c** and **7a**–**c**, involves two steps. In the first step, an N-alkylation of the benzo[f]quinoline with bromoketones **3a**–**c** take place and benzo[f]quinolinium bromides are obtained as previously reported [34]. The following step is a 3+2 dipolar cycloaddition of benzo[f]quinolinium ylides **4a**–**c** (generated in situ from the corresponding salts **3a**–**c**, in the presence of 1,2-butylene oxide as a catalyst) to the alkyne dipolarophiles (dimethyl acetylenedicarboxylate–DMAD or methyl propiolate), which are leading to final products, the cycloadducts **5a**–**c**, **7a**–**c**, **8c**.

The cycloaddition reactions were completed after 48 h of reflux. In the case of reactions of all salts with methyl propiolate, and in the case of the reactions of **3a,b** salts with DMAD only the fully aromatized compounds **5a**–**c** and **7a**–**b,** respectively, were obtained. In the case of the reaction of **3c** salt with DMAD, a mixture of fully aromatized compound **7c** and decarbonylated derivative **8c** was obtained.

The most probable reaction mechanism which explains the formation of these adducts is depicted in Scheme 2.

This mechanism involves, in the first step, the nucleophilic attack of the bromide ion on the 1,2-butylene oxide, the oxirane ring is opened, the alkoxide ion extracts a proton from the methylene group and leads to the ylide in 1,2-dipole form **4a**–**c**. The 1,2-dipole form adopts the 1,3-dipole **4a**–**c** form, which gives a Huisgen 3+2 cycloaddition with dipolarophiles via a concerted mechanism. The obtained reaction products **6a**–**c** are unstable (and non-isolable) because are non-aromatized, suffering a rearrangement to the more thermodynamic stable adducts **6a**–**c** (since the double bond in the pyrrole ring is in a conjugated system). The intermediate **6a**–**c** can be aromatized either by an oxidative dehydrogenation to a fully aromatized cycloadduct **7a**–**c**, or by a pivalaldehyde elimination in the case of the intermediate **6c** to a fully aromatized and decarbonylated derivative **8c**. The formation of the compound **8c** could be explained by the steric hindrance of the *tert*-butyl group from the **6c** intermediates. The partially aromatized derivatives **6a**–**c** are unstable when isolated, but were observed in the NMR spectra of the reaction mixture of all three benzo[f]quinolinium ylides **4a**–**c**. In the case of the benzo[f]quinolinium ylide **4a,** we have studied the time conversion of the **6a** intermediate (isolated after 24 h of reflux) to the final cycloadduct **7a** (see Appendix A). As can be observed from this study, the conversion of the intermediate **6a** to final cycloadduct **7a** takes several days in the absence of the 1,2-butylene oxide catalyst. If the reaction reflux occurs in 48 h, the amount of the intermediate **6a**–**c** is negligible. In addition, this study is solid prove for the proposed mechanism.

The final azasteroids were obtained in moderate to good yield (40 to 69% see Table 1). The long reaction time (48 h) and the high energy consumption are major disadvantages of the synthesis carried out under conventional conditions.

As a more energy-efficient alternative, we have performed the synthesis of the azasteroid derivatives under US irradiation, using a Bandelin reactor (Sonopuls GM 3200, Berlin, Germany), with a nominal power of 200 W, and an operating frequency of 20 kHz. The used reactor allowed us to control the pulse sequence, as well as the amplitude (mean percent of the nominal power) and the irradiation time. All these parameters are expected to influence the reaction. The cycloaddition reactions were performed using 80% of the instrument nominal power and were completed after 2 h of irradiation.

The data from Table 1 shows that under US irradiation the reaction times decrease substantially from 2 days to 2 h. At the same time, the solvent amounts used in the former were at least three times lower than the corresponding quantities used under conventional conditions (see Experimental). This qualifies the former reactions as environmentally friendly. We may also notice that under US irradiation the yields were slightly higher (by 12 to 27%). In the case of the reaction of bromide **3c** with DMAD, under US irradiation, we isolated less decarbonylated cycloadduct **8c** which makes this reaction more selective to the fully aromatized cycloadduct **7c**.

Optical properties of the synthesized azasteroids were investigated on diluted solutions (less than 10^−5^ mol/L) prepared in cyclohexane and trichloromethane, respectively. The dilution of each solution was adjusted, thus the absorbances maxima were measured in the 340–440 nm range, and reported on a 10 mm cuvette, to fit in a 0.5–0.9 unit range.

Since the compounds **5a**–**c, 7a**–**c**, and **8c** have a relatively similar structure, they exhibit small differences in their experimental UV-Vis absorption spectra, as can be seen from Figure 1 and Figure 2.

The absorption maxima of the seven azasteroid derivatives in cyclohexane and in trichloromethane are summarized in Table 2.

As we can observe in Table 2 and Figure 1 and Figure 2, the position of the absorption bands shows the influence of the substituents on the pyrrole ring.

In the case of all compounds (Figure 1 and Figure 2), in both solvents, three relatively well-separated absorption regions are observed (the first between 240–280 nm, the second between 280–325 nm, and the third between 325–425 nm, respectively). The absorption bands from the first region have a higher (double or more) intensity than the absorption bands from the second region, and the latter ones have a higher (less than double) intensity than the absorption bands from the third region. The absorption bands responsible for the blue emission of the azasteroids are situated in the third region (325–425 nm). In this region, four more or less separated absorption bands are visible. In cyclohexane, the four absorption bands are better separated than in trichloromethane.

In the case of the samples **5a**–**c**, the profile of the third absorption region and the absorption maxima are similar for the sample **5a** and **5b** (402 nm for **5a** and 403 nm for **5b** in trichloromethane, and 399 nm for both **5a** and **5b** in cyclohexane), while for the sample **5c** the maximum of the absorption is situated on 386 nm in trichloromethane and 384 nm in cyclohexane. Sample **5c** presents the same absorption bands as **5a** and **5b** with a small bathochromic shift.

In the case of the samples **7a**–**c** and **8c**, the absorption bands have similar intensities and profiles. In the case of the samples **7a** and **7b**, the absorption bands are not so well separated in trichloromethane but well separated in cyclohexane, while for the samples **7c** and **8c** these bands are separated both in trichloromethane and in cyclohexane. The maximum of the absorption of the sample **7a** presents a small bathochromic shift in comparison with the sample **7b**–**c** and **8c**. The maxima of the absorption are 381 nm for **7a**, 375 nm for **7b**, 370 nm for **7c**, and 369 nm for **8c** in trichloromethane, and 378 nm for **7a**, 376 nm for **7b**, 373 nm for **7c**, and 374 nm for **8c** in cyclohexane.

All the studied azasteroid derivatives have emission spectra consisting of one structured band in the 400–500 nm region (Figure 3, Figure 4, and Figure 5), indicating a planar structure of the molecules. The position of the band is significantly influenced by the presence of a carbomethoxy group at the 2nd position of the azasteroid skeleton. A hypsochromic shift of Δ_max_ = 7 nm (**5a** compared with **7a**), of Δ_max_ = 7 nm (**5b** compared with **7b**) and of Δ_max_ = 15 nm (**5c** compared with **7c**), could be observed in trichloromethane (Table 2). The same hypsochromic shift of Δ_max_ = 2 nm (**5a** compared with **7a**), of Δ_max_ = 2 nm (**5b** compared with **7b**), and of Δ_max_ = 3 nm (**5c** compared with **7c**), could be observed in cyclohexane (Table 2 and Figure 3).

When the substituent from the 3rd position of the azasteroid skeleton is the bulky pivaloyl group, an increase in the fluorescence intensity can be observed. A possible explanation can be the fact that the bulky group determines the deviation of the ketone group out of the molecule’s plane which breaks the conjugation between C=O carbonyl and the rest of the molecule, thus, the quenching of the fluorescence by this carbonyl group is reduced. When this carbonyl group from the 3rd position is missing (sample **8c**) the fluorescence intensity is maximum.

Figure 3 illustrate the absorption and emission spectra of compounds **5c, 7c**, and **8c** in trichloromethane (left column) and in cyclohexane (right column), the absorption and emission spectra of the other compounds (**5a,b** and **7a,b**) are presented in Appendix A.

The optical absorption and emission maxima of the azasteroid derivatives in cyclohexane and trichloromethane are summarized in Table 2.

Table 2, Figure 4 and Figure 5 show that all the compounds are blue emitters (λ_max_ of fluorescence around 430–450 nm). Compounds **7c** (in cyclohexane) and **8c** (both in trichloromethane and cyclohexane) have higher intensity in the emission spectra. Different behavior (in terms of the maximum of the emission) of the samples **5a,b** comparing with **5c**, and of the samples **7a,b** comparing with **7c** shown in Table 2 should relate to the difference in the electronic structures of **5a,b** and **7a,b** related to **5c** and **7c**, respectively. In the case of the compounds **5a,b** and **7a,b**, the substituent from the C=O keto group is methyl and ethyl, respectively, the difference between them is negligible, while in the case of the compounds **5c** and **7c** the substituent from the C=O keto group is *tert*-butyl, a bulkier one.

## 3. Experimental Section 

### 3.1. General Procedure

All the reagents and solvents were purchased from commercial sources and used without further purification except bromoacetone which was synthesized by the reaction of acetone with bromine in acetic acid as the catalyst. Melting points were recorded on an Electrothermal MEL-TEMP (Barnstead International, Dubuque, IA, USA) apparatus in open capillary tubes and are uncorrected. Analytical thin-layer chromatography was performed with commercial silica gel plates 60 F254 (Merck KGaA, Darmstadt, Germany) and visualized with UV light. The NMR spectra were recorded on a (Bruker Vienna, Wien Austria) Avance III 500 MHz spectrometer operating at 500 MHz for ^1^H and 125 MHz for ^13^C. The following abbreviations were used to designate chemical shift multiplicities: s = singlet, d = doublet, t = triplet, m = multiplet. Chemical shifts were reported in delta (δ) units, part per million (ppm), and coupling constants (*J*) in Hz. Infrared (IR) data were recorded in powder on the diamond crystal ATR mode (Pike Technologies, Fitchburg, MA, USA) on an FT-IR Vertex-70 (Bruker Optik, Leipzig, Germany) spectrophotometer. Ultrasound-assisted reactions were carried out using a Bandelin Ultrasound reactor (Sonopuls GM 3200, Berlin, Germany), with a nominal power of 200 W and a frequency of 20 kHz. The booster horn SH 213 G was fixed tightly to the ultrasonic converter. The titanium flat probe tip TT13 (diameter: 12.7 mm; length: 7 mm) was fixed tightly to the booster horn. The titanium probe tip was immersed in the used solvent. UV-Vis spectra were recorded on an (Shimadzu, Kyoto, Japan) 1800 PC spectrophotometer in cyclohexane and trichloromethane (spectroscopic grade) solution. The fluorescence measurements were made using an (Edinburgh Instruments, Livingstone, UK) F900 photoluminescence spectrometer, in the same solvents as for the UV-Vis spectra, with the excitation wavelength set to the absorption band maximum. For all spectral determinations, the solutions were kept in 10 mm path length quartz cells. The fluorescence quantum yield was determined at room temperature with an (Edinburgh Instruments, Livingstone, UK).

#### 3.1.1. General Procedure for Synthesis of Azasteroids Derivatives, **5a**–**b**, **7a**–**b** and **8c** under Conventional TH Conditions

A mixture of benzo[f]quinolinium salt **3a**–**c** (0.791 g, 2.5 mmol of **3a** or 0,826 g, 2.5 mmol of **3b** or 0.896 g, 2.5 mmol of **3c**) and methyl propiolate (0.31 mL, 3.5 mmol) or dimethyl acetylenedicarboxylate (0.43 mL, 3.5 mmol) was suspended in 30 mL 1,2-butylene oxide. The stirring and refluxing were continued for 48 h. After the reaction was finished (TLC), the obtained solution was cooled down at room temperature and evaporated under reduced pressure to give the crude product. The purification of the crude product was done by column chromatography on silica gel (eluted with 99.5/0.5 CH_2_Cl_2_/CH_3_OH).

#### 3.1.2. General Procedure for Synthesis of Azasteroids Derivatives, **5a**–**b**, **7a**–**b** and **8c** under US Irradiation

Under US irradiation, the mixture of reagents benzo[f]quinolinium salt **3a**–**c** (0.791 g, 2.5 mmol of **3a** or 0,826 g, 2.5 mmol of **3b** or 0.896 g, 2.5 mmol of **3c**) and methyl propiolate (0.31 mL, 3.5 mmol) or dimethyl acetylenedicarboxylate (0.43 mL, 3.5 mmol) in 10 mL 1,2-butylene oxide was placed into the reaction vessel and exposed to irradiation (from 2 h; see Table 1). Once the irradiation cycle was completed, the reaction tube was removed from the reactor, and processed as indicated above for TH condition.

*Methyl 3-acetylbenzo[f]pyrrolo[1,2-a]quinoline-1-carboxylate* (**5a**). (0.516 g, 65% (under classical heating) and 0.635 g, 80% (under ultrasounds)) as a straw-yellow crystals, m.p. 215–216 °C); *R*_f_ (99.5/0.5 CH_2_Cl_2_/CH_3_OH) 0.44; IR (cm^−1^): 3034, 3013 (C-H arom.), 2963, 2927 (C-H aliph.), 1704 (C=O, ester), 1639 (C=O, keto), 1610, 1547, 1497, 1442, 1415, 1406, 1353 (aromatic and heteroaromatic ring), 1243, 1178, 1157, 1098, 1064 (C–O–C, ester); ^1^H NMR (500 MHz, CDCl_3_): δ 8.56 (2H, m, overlapped peaks, H-10, H-11), 8.47 (1H, d, *J* = 9.5 Hz, H-12), 8.10 (2H, m, overlapped peaks, H-2, H-5), 7.96 (2H, m, overlapped peaks, H-6, H-7), 7.71 (1H, t, *J* = 7.5 Hz, H-9), 7.63 (1H, t, *J* = 7.5 Hz, H-8), 3.97 (3H, s, CH_3_ of methoxycarbonyl group from 1st position), 2.75 (3H, s, CH_3_ of acetyl group from 3rd position); ^13^C NMR (125 MHz, CDCl_3_): δ 186.8 (CO keto group from 3^rd^ position), 164.6 (CO keto ester from 1^st^ position), 140.5 (C-12a), 132.5 (C-4a), 131.0 (C-6a), 129.7 (C-10a), 129.1 (C-6), 128.9 (C-7), 128.6 (C-5), 128.6 (C-3), 127.7 (C-9), 126.8 (C-8), 124.2 (C-11), 123.0 (C-10), 121.2 (C-10b), 120.4 (C-2), 117.5 (C-12), 106.7 (C-1), 51.6 (CH_3_ of methoxycarbonyl group from 1st position), 28.4 (CH_3_ of acetyl group from 3rd position). Anal. calc. for C_20_H_15_NO_3_ (317.34): C 75.70, H 4.76, N 4.41; found: C 75.65, H 4.70, N 4.35.

*Methyl 3-propionylbenzo[f]pyrrolo[1,2-a]quinoline-1-carboxylate* (**5b**). (0.572 g, 69% (under classical heating) and 0.671 g, 81% (under ultrasounds)) as a straw-yellow crystals, m.p. 186–187 °C); *R*_f_ (99.5/0.5 CH_2_Cl_2_/CH_3_OH) 0.58; IR (cm^−1^): 3034, 3001 (C-H arom.), 2978, 2940 (C-H aliph.), 1705 (C=O, ester), 1645 (C=O, keto), 1610, 1547, 1496, 1442, 1414, 1359 (aromatic and heteroaromatic ring), 1238, 1180, 1156, 1128, 1088, 1073 (C–O–C, ester); ^1^H NMR (500 MHz, CDCl_3_): δ 8.56 (1H, d, *J* = 8.5 Hz, H-10), 8.54 (1H, d, *J* = 9.5 Hz, H-11), 8.46 (1H, d, *J* = 9.5 Hz, H-12), 8.08 (1H, s, H-2), 8.01 (1H, d, *J* = 9.0 Hz, H-5), 7.96 (2H, m, overlapped peaks, H-6, H-7), 7.70 (1H, t, *J* = 7.0, 8.5 Hz, H-9), 7.62 (1H, t, *J* = 7.0 Hz, H-8), 3.97 (3H, s, CH_3_ of methoxycarbonyl group from 1st position), 3.10 (2H, q, *J* = 7.5 Hz, CH_2_ of propionyl group from 3^rd^ position), 1.38 (3H, t, *J* = 7.5 Hz, CH_3_ of propionyl group from 3rd position); ^13^C NMR (125 MHz, CDCl_3_): δ 190.9 (CO keto group from 3^rd^ position), 164.6 (CO keto ester from 1st position), 140.3 (C-12a), 132.5 (C-4a), 131.0 (C-6a), 129.7 (C-10a), 129.1 (C-6), 128.8 (C-7), 128.6 (C-3), 127.7 (C-8), 127.6 (C-2), 126.8 (C-9), 123.9 (C-11), 123.0 (C-10), 121.2 (C-10b), 120.3 (C-5), 117.6 (C-12), 106.7 (C-1), 51.5 (CH_3_ of methoxycarbonyl group from 1st position), 33.7 (CH_2_ of propionyl group from 3^rd^ position), 9.9 (CH_3_ of propionyl group from 3rd position). Anal. calc. for C_21_H_17_NO_3_ (331.37): C 76.12, H 5.17, N 4.23; found: C 76.07, H 5.13, N 4.17.

*Methyl 3-pivaloylbenzo[f]pyrrolo[1,2-a]quinoline-1-carboxylate* (**5c**). (0.575 g, 64% (under classical heating) and 0.701 g, 78% (under ultrasounds)) as a straw-yellow crystals, m.p. 192–193 °C); *R*_f_ (99.5/0.5 CH_2_Cl_2_/CH_3_OH) 0.67; IR (cm^−1^): 3057 (C-H arom.), 2982, 2950, 2930 (C-H aliph.), 1700 (C=O, ester), 1648 (C=O, keto), 1611, 1545, 1507, 1493, 1444, 1415, 1367 (aromatic and heteroaromatic ring), 1235, 1152, 1138, 1119, 1084 (C–O–C, ester); ^1^H NMR (500 MHz, CDCl_3_): δ 8.50 (1H, d, *J* = 8.0 Hz, H-10), 8.40 (2H, m, overlapped peaks, H-11, H-12), 7.92 (1H, s, H-2), 7.90 (2H, m, overlapped peaks, H-5, H-7), 7.65 (1H, t, *J* = 7.5, 8.0 Hz, H-9), 7.57 (1H, t, *J* = 7.5 Hz, H-8), 7.54 (1H, d, *J* = 9.0 Hz, H-6), 3.96 (3H, s, CH_3_ of methoxycarbonyl group from 1st position), 1.58 (9H, s, 3xCH_3_ of *tert*-butyl group from 3rd position); ^13^C NMR (125 MHz, CDCl_3_): δ 199.7 (CO keto group from 3rd position), 164.8 (CO keto ester from 1st position), 138.9 (C-12a), 131.9 (C-4a), 130.6 (C-6a), 129.9 (C-10a), 129.3 (C-5), 128.8 (C-7), 127.7 (C-9), 126.9 (C-3), 126.6 (C-8), 124.1 (C-2), 122.9 (C-10), 122.8 (C-11), 120.8 (C-10b), 119.2 (C-6), 117.6 (C-12), 105.9 (C-1), 51.4 (CH_3_ of methoxycarbonyl group from 1st position), 44.4 (C of *tert*-butyl group from 3rd position) 29.1 (3xCH_3_ of *tert*-butyl group from 3rd position). Anal. calc. for C_23_H_21_NO_3_ (359.43): C 76.86, H 5.89, N 3.90; found: C 76.80, H 5.86, N 3.85.

*Dimethyl 3-acetylbenzo[f]pyrrolo[1,2-a]quinoline-1,2-dicarboxylate* (**7a**). (0.507 g, 54% (under classical heating) and 0.666 g, 71% (under ultrasounds)) as a straw-yellow crystals, m.p. 197–198 °C); *R*_f_ (99.5/0.5 CH_2_Cl_2_/CH_3_OH) 0.26; IR (cm^−1^): 3067, 3003 (C-H arom.), 2952 (C-H aliph.), 1739, 1695 (C=O, ester), 1662 (C=O, keto), 1613, 1547, 1491, 1468, 1444, 1405 (aromatic and heteroaromatic ring), 1247, 1219, 1197, 1166, 1154, 1096 (C–O–C, ester); ^1^H NMR (500 MHz, CDCl_3_): δ 8.55 (1H, d, *J* = 8.5 Hz, H-10), 8.51 (1H, d, *J* = 9.5 Hz, H-11), 8.40 (1H, d, *J* = 9.5 Hz, H-12), 7.95 (2H, d, overlapped peaks, H-5, H-7), 7.73 (1H, t, *J* = 7.5 Hz, H-9), 7.64 (2H, m, overlapped peaks, H-6, H-8), 4.05 (3H, s, CH_3_ of methoxycarbonyl group from 1st position), 3.95 (3H, s, CH_3_ of methoxycarbonyl group from 2nd position), 2.60 (3H, s, CH_3_ of acetyl group from 3rd position); 13C NMR (125 MHz, CDCl_3_): δ 189.0 (CO keto group from 3rd position), 167.0 (CO keto ester from 2nd position), 163.6 (CO keto ester from 1st position), 137.8 (C-12a), 131.7 (C-4a), 130.9 (C-6a), 130.5 (C-10a), 129.7 (C-5), 129.7 (C-2), 128.9 (C-7), 128.1 (C-9), 127.1 (C-8), 126.9 (C-3), 123.9 (C-11), 123.0 (C-10), 121.5 (C-10b), 119.3 (C-6), 117.9 (C-12), 104.9 (C-1), 53.4 (CH_3_ of methoxycarbonyl group from 1st position), 52.0 (CH_3_ of methoxycarbonyl group from 2nd position), 29.5 (CH_3_ of acetyl group from 3rd position). Anal. calc. for C_22_H_17_NO_5_ (375.38): C 70.39, H 4.56, N 3.73; found: C 70.33, H 4.52, N 3.69.

*Dimethyl 3-propionylbenzo[f]pyrrolo[1,2-a]quinoline-1,2-dicarboxylate* (**7b**). (0.526 g, 54% (under classical heating) and 0.672 g, 69% (under ultrasounds)) as a straw-yellow crystals, m.p. 206–207 °C); *R*_f_ (99.5/0.5 CH_2_Cl_2_/CH_3_OH) 0.30; IR (cm^−1^): 3032, 3000 (C-H arom.), 2981, 2951 (C-H aliph.), 1731, 1697 (C=O, ester), 1668 (C=O, keto), 1612, 1546, 1494, 1467, 1444, 1406, 1364 (aromatic and heteroaromatic ring), 1266, 1223, 1182, 1163, 1145, 1095, 1089 (C–O–C, ester); ^1^H NMR (500 MHz, CDCl_3_): δ 8.54 (1H, d, *J* = 8.5 Hz, H-10), 8.46 (1H, d, *J* = 9.5 Hz, H-11), 8.38 (1H, d, *J* = 9.5 Hz, H-12), 7.95 (2H, d, overlapped peaks, H-5, H-7), 7.72 (1H, t, *J* = 7.5 Hz, H-9), 7.63 (1H, t, *J* = 7.5 Hz, H-8), 7.56 (1H, d, *J* = 9.5 Hz, H-6), 4.03 (3H, s, CH_3_ of methoxycarbonyl group from 1st position), 3.94 (3H, s, CH_3_ of methoxycarbonyl group from 2^nd^ position), 2.85 (2H, q, *J* = 7.5 Hz, CH_2_ of propionyl group from 3rd position), 1.31 (3H, t, *J* = 7.5 Hz, CH_3_ of propionyl group from 3rd position); ^13^C NMR (125 MHz, CDCl_3_): δ 194.3 (CO keto group from 3rd position), 166.7 (CO keto ester from 2nd position), 163.7 (CO keto ester from 1st position), 137.2 (C-12a), 131.5 (C-4a), 130.8 (C-6a), 129.8 (C-5), 129.7 (C-10a), 128.9 (C-7), 128.8 (C-2), 128.1 (C-9), 127.1 (C-8), 127.0 (C-3), 123.2 (C-11), 123.0 (C-10), 121.4 (C-10b), 118.7 (C-6), 118.1 (C-12), 104.7 (C-1), 53.3 (CH_3_ of methoxycarbonyl group from 1st position), 51.9 (CH_3_ of methoxycarbonyl group from 2nd position), 35.6 (CH_2_ of propionyl group from 3rd position), 9.1 (CH_3_ of propionyl group from 3rd position). Anal. calc. for C_23_H_19_NO_5_ (389.41): C 70.94, H 4.92, N 3.60; found: C 70.87, H 4.88, N 3.55.

*Dimethyl 3-pivaloylbenzo[f]pyrrolo[1,2-a]quinoline-1,2-dicarboxylate* (**7c**). (0.397 g, 38% (under classical heating) and 0.678 g, 65% (under ultrasounds)) as a straw-yellow crystals, m.p. 184–185 °C); *R*_f_ (99.5/0.5 CH_2_Cl_2_/CH_3_OH) 0.35; IR (cm^−1^): 3067, 3023 (C-H arom.), 2981, 2971, 2953 (C-H aliph.), 1729, 1694 (C=O, ester), 1682 (C=O, keto), 1612, 1559, 1546, 1505, 1464, 1418 (aromatic and heteroaromatic ring), 1258, 1208, 1177, 1161, 1119, 1091, 1059 (C–O–C, ester); ^1^H NMR (500 MHz, CDCl_3_): δ 8.56 (1H, d, *J* = 8.0 Hz, H-10), 8.35 (2H, m, overlapped peaks, H-11, H-12), 7.95 (2H, m, overlapped peaks, H-5, H-7), 7.73 (2H, m, overlapped peaks, H-6, H-9), 7.64 (1H, t, *J* = 7.0 Hz, H-8), 3.95 (3H, s, CH_3_ of methoxycarbonyl group from 1st position), 3.93 (3H, s, CH_3_ of methoxycarbonyl group from 2nd position), 1.15 (9H, s, 3xCH_3_ of *tert*-butyl group from 3rd position); ^13^C NMR (125 MHz, CDCl_3_): δ 207.9 (CO keto group from 3rd position), 165.6 (CO keto ester from 2nd position), 164.0 (CO keto ester from 1st position), 134.4 (C-12a), 131.8 (C-4a), 130.8 (C-6a), 130.1 (C-5), 129.8 (C-10a), 128.9 (C-7), 128.3 (C-9), 127.0 (C-8), 126.4 (C-2), 123.0 (C-10), 122.9 (C-3), 120.9 (C-11), 120.7 (C-10b), 118.7 (C-12), 116.9 (C-6), 104.6 (C-1), 52.8 (CH_3_ of methoxycarbonyl group from 1st position), 51.9 (CH_3_ of methoxycarbonyl group from 2nd position), 47.2 (C of *tert*-butyl group from 3rd position), 27.4 (3xCH_3_ of *tert*-butyl group from 3rd position). Anal. calc. for C_25_H_23_NO_5_ (417.16): C 71.93, H 5.55, N 3.36; found: C 71.88, H 5.50, N 3.32.

*Dimethyl benzo[f]pyrrolo[1,2-a]quinoline-1,2-dicarboxylate* (8c). (0.108 g, 13% (under classical heating) and 0.042 g, 5% (under ultrasounds)) as a straw-yellow crystals, m.p. 198–199 °C); *R*_f_ (99.5/0.5 CH_2_Cl_2_/CH_3_OH) 0.28; IR (cm^−1^): 3145, 3091, 3066 (C-H arom.), 2982, 2963, 2942 (C-H aliph.), 1710, 1675 (C=O, ester), 1610, 1563, 1545, 1507, 1463, 1441, 1406 1369 (aromatic and heteroaromatic ring), 1270, 1255, 1231, 1217, 1197, 1164 1095, 1060 (C–O–C, ester); ^1^H NMR (500 MHz, CDCl_3_): δ 8.41 (1H, d, *J* = 8.5 Hz, H-10), 8.21 (1H, s, H-3), 8.11 (1H, d, *J* = 9.5 Hz, H-11), 8.06 (1H, d, *J* = 9.5 Hz, H-12), 7.88 (3H, m, overlapped peaks, H-5, H-6, H-7), 7.66 (1H, t, *J* = 8.0 Hz, H-9), 7.56 (1H, t, *J* = 8.0 Hz, H-8), 3.97 (3H, s, CH_3_ of methoxycarbonyl group from 1st position), 3.96 (3H, s, CH_3_ of methoxycarbonyl group from 2nd position); ^13^C NMR (125 MHz, CDCl_3_): δ 165.0 (CO keto ester from 2nd position), 164.5 (CO keto ester from 1st position), 134.9 (C-12a), 130.8 (C-6a), 130.5 (C-6), 130.1 (C-4a), 129.8 (C-10a), 129.0 (C-7), 128.0 (C-9), 126.5 (C-8), 122.7 (C-10), 120.7 (C-2), 119.5 (C-11), 119.3 (C-10b), 118.6 (C-12), 116.1 (C-3), 113.9 (C-5), 105.4 (C-1), 52.3 (CH_3_ of methoxycarbonyl group from 1st position), 51.6 (CH_3_ of methoxycarbonyl group from 2nd position). Anal. calc. for C_20_H_15_NO_4_ (333.34): C 72.06, H 4.54, N 4.20; found: C 72.00, H 4.49, N 4.16.

## 4. Conclusions

In conclusion, we report herein an efficient and straightforward pathway for obtaining a new class of blue fluorescent azasteroid derivatives, under conventional (thermal) heating as well as under unconventional (ultrasound) irradiation. Blue fluorescent azasteroid derivatives have been obtained using a cycloaddition reaction of benzo[f]quinolinium ylides with symmetrically and unsymmetrically activated alkynes. Under conventional heating, in the case of reactions with methyl propiolate only the fully aromatized compounds **5a**–**c** were obtained, while in the case of reactions with DMAD fully aromatized compounds **7a**–**b** and a mixture of fully aromatized compound **7c** and a fully aromatized and decarbonylated derivative **8c** were obtained. A feasible reaction mechanism for the azasteroid derivatives formation is described in this study. Under ultrasound irradiation, we isolated fully aromatized cycloadducts with increased yield, and in the case of the reaction of bromide **3c** with DMAD, we isolated less decarbonylated cycloadduct **8c** which make this reaction more selective to the fully aromatized cycloadduct **7c**.

The absorption and emission maxima of the obtained azasteroid derivatives were studied, some of these compounds being blue emitters. The absorption and emission spectra are dependent on their structure, compounds with a similar structure having spectra with small differences, whereas the compounds with a bulky pivaloyl group or without a C=O keto group in the 3rd position present more intense fluorescence emission.

We also note that under ultrasound irradiation, the reactions occur with increased selectivity regarding the decarbonylated compound and offers several advantages in terms of yield, easier workup of the reaction, a substantial decrease in consumed solvents, a substantial reduction in reaction time (from days to hours), thus, consequent diminution in energy consumption. Taking into consideration these advantages, the proposed method should be considered environmentally friendly.

## Data Availability

Not available.

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
