# Peer review of "Fluorescent Azasteroids through Ultrasound Assisted Cycloaddition Reactions"

_molecules, 2021, doi:10.3390/molecules26165098_

Round 1

Reviewer 1 Report

The article "Fluorescent azasteroids through ultrasound assisted cycloaddition reactions" deals with the synthesis of unknown molecules of azasteroids by two mode of heating and their photophysical properties. 

The article is well written and the characterisation were well done, as the photophysical properties.

For me the article is good for publication.

Author Response

Thank you very much for your appreciation.

Reviewer 2 Report

The manuscript “Fluorescent azasteroids through ultrasound assisted cycloaddition reactions” by C. Moldoveanu and co-authors describes synthesis via ultrasound irradiation and photophysical properties of azasteroid compounds. It reported that under ultrasound irradiation, the yields of the azasteroids increased relative to conventional thermal heating condition. The data for compounds (1H NMR, IR etc.) are adequately solid. I recommend acceptance for publication after addressing the following points.

  1. The authors describe 1,2-butylene oxide as ‘catalyst’. However, the reagent is converted into corresponding alcohol by epoxy ring opening in possible mechanism. In addition, the reagent was used as a solvent in the reaction (not catalytic amount). I think the reagent should be represented as a ‘reactant’.
  2. Is the pivalaldehyde elimination in aromatization reaction recognized? If any examples have been reported, the authors should cite these. Furthermore, it dose not occur in the compound 5c. It is indefinite to explain the formation of 8c only by the steric hinderance.
  3. The authors should explain why the fluorescence intensity of 7c in trichloromethane decreased compared with that in cyclohexane. If solvent effect is involved, I am interested in the case of the other polar solvents.

Minor comments:

  • P2-L71; 5a-c should be 5a-b.
  • P3-L88; ould should be could?
  • P8-L219; dichloromethane should be trichloromethane.

Author Response

  1. The authors describe 1,2-butylene oxide as ‘catalyst’. However, the reagent is converted into corresponding alcohol by epoxy ring opening in possible mechanism. In addition, the reagent was used as a solvent in the reaction (not catalytic amount). I think the reagent should be represented as a ‘reactant’.

We modified “catalyst” to “reactant”. Thank you for the suggestion.

  1. Is the pivalaldehyde elimination in aromatization reaction recognized? If any examples have been reported, the authors should cite these. Furthermore, it dose not occur in the compound 5c. It is indefinite to explain the formation of 8c only by the steric hinderance.

We did not find example in the literature where pivalaldehyde is eliminated in an aromatization reaction, but we found different molecule elimination in order to obtain aromatic ring after cycloaddition reaction (see reference 25 from the manuscript).

Regarding the fact that in the case of the compound 5c this elimination does not occur, in this compound (5c) the 2nd position of the azasteroid skeleton is occupied by a hydrogen atom (the smallest atom) while in the case of the compound 7c, the 2nd position of the azasteroid skeleton is occupied by the bulkier carbomethoxy group. We think that you will admit that the steric hinderance with pivaloyl group from the 3rd position in the last case is many times stronger than in the first case. That is explanation of the pivalaldehyde elimination in the last case, while in the first case this elimination does not occur.

  1. The authors should explain why the fluorescence intensity of 7c in trichloromethane decreased compared with that in cyclohexane. If solvent effect is involved, I am interested in the case of the other polar solvents.

In our fluorescence study (see references 16 and 17) we observed the same behavior regarding the fluorescence of the studied compounds, namely that the fluorescence intensity is higher in non-polar solvents than in polar solvents. This behavior may be explained by the dipole-dipole interaction between the molecules of the solvent and the molecules of the fluorescent compounds. Fluorescence require a planar structure of the molecule. The interaction between solvent and the fluorescent molecules may modify the planar structure of the molecule. The greater are these interaction (the solvent is more polar), the more affected will be the structure of the molecule and the intensity of the fluorescence will decrease. This behavior is more obvious in the case of the compound 7c since the dipole moment of the pivaloyl group is bigger than the dipole moment of the methyl or ethyl group (from the compound 7a and 7b respectively) and his interaction with trichloromethane molecule is stronger. This stronger interaction makes the deformation of the planar structure of the molecule 7c been greater and the fluorescence intensity smaller than in cyclohexane (a less polar solvent).

Minor comments:

  • P2-L71; 5a-c should be 5a-b.

Not, in the case of the reaction of all three salts with methyl propiolate we have obtained only the fully aromatized compounds. In the case of the salt 3c we did not obtained the decarbonylated compound in the reaction with methyl propiolate. Thank you for your remark.

  • P3-L88; ould should be could?

Done. Thank you for your remark.

  • P8-L219; dichloromethane should be trichloromethane.

Done. Thank you for your remark.